# Omega-3 Fatty Acid-Enriched Fish Oil and Selenium Combination Modulates Endoplasmic Reticulum Stress Response Elements and Reverses Acquired Gefitinib Resistance in HCC827 Lung Adenocarcinoma Cells

**DOI:** 10.3390/md18080399

**Published:** 2020-07-29

**Authors:** Chien-Huang Liao, Yu-Tien Tzeng, Gi-Ming Lai, Chia-Lun Chang, Ming-Hung Hu, Wei-Lun Tsai, Yun-Ru Liu, Simon Hsia, Shuang-En Chuang, Tzeon-Jye Chiou, Le-Ming Wang, Jacqueline Whang-Peng, Chih-Jung Yao

**Affiliations:** 1Cancer Center, Wan Fang Hospital, Taipei Medical University, Taipei 11696, Taiwan; a2639264@ms25.hinet.net (C.-H.L.); gminlai@canceraway.org.tw (G.-M.L.); clairgg@hotmail.com (W.-L.T.); 108178@w.tmu.edu.tw (T.-J.C.); jqwpeng@nhri.org.tw (J.W.-P.); 2Division of Pulmonary Medicine, Wan Fang Hospital, Taipei Medical University, Taipei 11696, Taiwan; yttzeng7@gmail.com; 3Division of Hematology and Medical Oncology, Department of Internal Medicine, Wan Fang Hospital, Taipei Medical University, Taipei 11696, Taiwan; 101255@w.tmu.edu.tw (C.-L.C.); lancehu7@gmail.com (M.-H.H.); 4Department of Internal Medicine, School of Medicine, College of Medicine, Taipei Medical University, Taipei 11031, Taiwan; 5Joint Biobank, Office of Human Research, Taipei Medical University, Taipei 11031, Taiwan; d90444002@tmu.edu.tw; 6Taiwan Nutraceutical Association, Taipei 10596, Taiwan; Dr.Simon.hsia@gmail.com; 7National Institute of Cancer Research, National Health Research Institutes, Miaoli 35053, Taiwan; sechuang@nhri.org.tw; 8Department of Obstetrics and Gynecology, Wan Fang Hospital, Taipei Medical University, Taipei 11696, Taiwan; lemingmd90@gmail.com

**Keywords:** lung cancer, fish oil, selenium, gefitinib, acquired resistance

## Abstract

Non-small cell lung cancer (NSCLC)-carrying specific epidermal growth factor receptor (EGFR) mutations can be effectively treated by a tyrosine kinase inhibitor such as gefitinib. However, the inevitable development of acquired resistance leads to the eventual failure of therapy. In this study, we show the combination effect of omega-3 fatty acid-enriched fish oil (FO) and selenium (Se) on reversing the acquired gefitinib-resistance of HCC827 NSCLC cells. The gefitinib-resistant subline HCC827GR possesses lowered proapoptotic CHOP (CCAAT/enhancer-binding protein homologous protein) and elevated cytoprotective GRP78 (glucose regulated protein of a 78 kDa molecular weight) endoplasmic reticulum (ER) stress response elements, and it has elevated β-catenin and cyclooxygenase-2 (COX-2) levels. Combining FO and Se counteracts the above features of HCC827GR cells, accompanied by the suppression of their raised epithelial-to-mesenchymal transition (EMT) and cancer stem markers, such as vimentin, AXL, N-cadherin, CD133, CD44, and ABCG2. Accordingly, an FO and Se combination augments the gefitinib-mediated growth inhibition and apoptosis of HCC827GR cells, along with the enhanced activation of caspase -3, -9, and ER stress-related caspase-4. Intriguingly, gefitinib further increases the elevated ABCG2 and cancer stem-like side population in HCC827GR cells, which can also be diminished by the FO and Se combination. The results suggest the potential of combining FO and Se in relieving the acquired resistance of NSCLC patients to targeted therapy.

## 1. Introduction

Non-small cell lung cancer (NSCLC) constitutes more than 80% of lung cancers, which is the leading cause of cancer death worldwide [1,2]. A subset of NSCLC-carrying specific genetic mutations of epidermal growth factor receptor (EGFR) can be effectively treated by EGFR tyrosine kinase inhibitor (TKI)-mediated targeted therapy. The two most commonly (>90%) characterized EGFR-activating mutations conferring sensitivity to EGFR-TKIs are small in-frame deletions in exon 19 (particularly E746-A750del, an in-frame deletion of five amino acids (glutamic acid, leucine, arginine, glutamic acid, and alanine) from position 746 to 750) and amino acid substitution in exon 21 (leucine to arginine at codon 858 (L858R)) [3,4]. First-generation EGFR-TKIs, such as gefitinib and erlotinib, have exhibited significant effects in prolonging the disease-free survival of NSCLC patients whose tumors harbor EGFR mutations [5,6]. However, most patients who initially respond to EGFR-TKIs eventually acquire resistance within 9–14 months [6], leading to the failure of therapy.

Various mechanisms have been discovered to be involved in the acquired resistance to EGFR-TKI, including secondary EGFR T790M (substitution of the putative gatekeeper threonine residue at position 790 with methionine within exon 20 of EGFR) and minor mutations, the amplification of tyrosine kinase receptors such as, Anexelekto (AXL), hepatocyte growth factor receptor (MET), and human epidermal growth factor receptor 2 (HER2) for bypass and alternative signaling pathways, small-cell lung cancer transformation, acquiring epithelial-to-mesenchymal transition (EMT) signatures and cancer stem cell traits, as well as additional mechanisms remain to be identified [5,6,7,8]. Aside from these well-known mechanisms, the modulation of endoplasmic reticulum (ER) stress response elements such as proapoptotic CHOP (CCAAT/enhancer-binding protein homologous protein, growth arrest and DNA-damage protein 153 (GADD153)) and prosurvival GRP78 (glucose-regulated protein of a 78 kDa molecular weight, binding immunoglobulin protein (BiP)) is involved in the gefitinib-mediated cytotoxicity of NSCLC cells [9,10,11]. The upregulation or knockdown of CHOP has been shown to enhance or inhibit EGFR-TKI-induced apoptosis, respectively, in NSCLC cells [9,10,11]. Combined treatment to switch the ER stress response from cytoprotective to proapoptotic signaling upon EGFR-TKI treatment might be an alternative approach for overcoming the acquired drug resistance that hampers the success of NSCLC-targeted therapy.

Fish oil (FO) is enriched of omega-3 fatty acids, particularly docosahexaenoic acid (DHA) and eicosapentaenoic acid (EPA) [12]. In addition to cyclooxygenase-2 (COX-2) suppression [13] and cardiovascular protection [14], omega-3 fatty acids also have effects against lung cancer cells [13,15,16]. Supplementation with fish oil increases first-line chemotherapy efficacy in advanced NSCLC patients [17]. Selenium (Se) is an essential trace element for human health, and its potential in cancer treatment and chemoprevention has attracted considerable attention [18,19]. Many aquatic species have been harvested as a source of Se-enriched biomass and marketed as healthy foods [20,21]. For example, the green alga *Ulva fasciata* is a good Se carrier and has been suggested to be a source of Se-enriched food [20]. The Se-containing complex in *Ulva fasciata* has been shown to induce mitochondria-mediated apoptosis in A549 human NSCLC cells [22]. According to Zhong et al. [20], most of the Se accumulated in *Ulva fasciata* is transformed into the organic form. Edible seaweeds have great potential to transform inorganic Se into organic forms by metabolic processes [21]. In general, it is believed that organic selenocompounds are better and safer than inorganic Se [20,23]. Organic Se is an important Se sources including Se amino acid, Se polysaccharide, and Se yeast [24]. Among them, Se yeast is produced by growing select strains of *Saccharomyces cerevisiae* in Se-rich media [25]. It predominantly contains l-selenomethionine [19] and has an excellent safety record [25]. In this study, Se yeast was used to treat NSCLC cells in order to investigate the combination effect of FO and Se.

Both omega-3 fatty acid and Se have been shown to exert their anticancer activities through the induction of ER stress-associated apoptosis in cancer cells [26,27,28,29,30]. Our previous study found the synergistic combination effect of FO omega-3 fatty acid and Se on the apoptosis induction of NSCLC cells through the opposite regulation of CHOP and GRP78 [31]. Moreover, the combination of FO and Se also suppresses β-catenin and COX-2 [31], of which overexpression is associated with gefitinib resistance of lung cancer cells [32,33]. These findings of our previous work suggest the potentiality of combining FO and Se to reverse the acquired resistance of NSCLC cells to EGFR-TKI through modulating ER stress response elements as aforementioned.

In the present study, we established a gefitinib-resistant subline (HCC827GR) from the gefitinib-sensitive human NSCLC cell line HCC827, which carries the canonical E746-A750 exon 19 deletion [34]. The ER stress response elements, such as CHOP and GRP78, as well as β-catenin and COX-2 levels, were compared between the HCC827GR and parental HCC827 cells, in addition to the markers for the well-known mechanisms mentioned above. At a clinically achievable concentration [35,36], we explored the combination effect of FO and Se on modulating the ER stress response elements in HCC827GR cells. The subsequent enhancement of the gefitinib-induced apoptosis and inhibition of the above-mentioned known markers related to EGFR-TKI resistance were examined.

## 2. Results

### 2.1. The Gefitinib-Resistant Subline HCC827GR Possesses Higher GRP78, β-Catenin, and COX-2 but Has Lower CHOP Than the Parental HCC827

To evaluate the combination effect of FO and Se on reversing the acquired resistance of NSCLC cells to EGFR-TKI such as gefitinib, a resistant subline HCC827GR derived from the gefitinib-sensitive HCC827 human NSCLC cell line was employed. The HCC827 cells were initially very sensitive to gefitinib. After treatment with a 0.125 μM concentration of gefitinib for 72 h, the viability of HCC827 cells was decreased to 20.8% of the control (Figure 1a, left panel). By contrast, the viability of HCC827GR cells was only decreased to 73.6% by the same concentration of gefitinib (Figure 1a, right panel). Even when the gefitinib concentration was increased to 1 μM, its inhibition on HCC827GR cell viability was almost the same as that by 0.125 μM (Figure 1a, right panel). It has been reported that the maximum plasma concentrations of gefitinib resulting from clinically relevant doses are 0.5–1 μM or more [37]. At a concentration of 1 μM, gefitinib caused 64.3% and 4.9% of apoptosis (sub-G1 fraction) in the parental HCC827 (Figure 1b, upper panel) and the resistant HCC827GR (Figure 1b, lower panel) cells, respectively, after 96 h of treatment. The selected HCC827GR subline appeared to have acquired a resistance to gefitinib.

In this pair of cell lines, we then analyzed their content of gefitinib resistance-related proteins that were expected to be regulated by the combination of FO and Se. As shown in Figure 2a, the gefitinib-resistant HCC827GR possessed lower proapoptotic CHOP but much higher prosurvival GRP78 ER stress response elements in comparison to parental HCC827. Moreover, the HCC827GR had much higher levels of β-catenin and COX-2, two known gefitinib resistance-associated proteins [32,33], than the parental HCC827 (Figure 2a), thus showing the phenotype of acquired gefitinib-resistance.

### 2.2. The EMT Markers and Cancer Stem Cell Traits Are Raised in Gefitinib-Resistant Subline HCC827GR

The contribution of the EMT to the acquired resistance of NSCLC cells to EGFR-TKI is well-documented [5,8,38], and the upregulation of the EMT marker vimentin may drive AXL overexpression [8]. In agreement, we found markedly higher vimentin and AXL levels accompanied by increased N-cadherin and decreased E-cadherin protein levels in HCC827GR compared with the parental HCC827 (Figure 2b). Consistent with those results described by Ware et al. [39], besides AXL, phospho-MET (p-MET) was also amplified in HCC827GR cells upon adaptation to gefitinib (Figure 2b).

Similar to that observed by Shien et al. [5], these HCC827GR cells also exhibited cancer stem cell traits, as evidenced by the raised lung cancer stem markers such as CD133, ATP-binding cassette super-family G member 2 (ABCG2), and CD44 (Figure 2c).

In accordance with the increase of ABCG2, a molecular determinant of the side-population phenotype [40], the proportion of cancer stem-like side population in the HCC827GR cells was raised to 2.53% from 0.3% of the parental HCC827 (Figure 3). In addition to the rise of prosurvival signals shown in Figure 2a, these HCC827GR cells acquired features of the EMT and cancer stem cells, as well as amplified bypass and alternative signaling pathways, all of which are consistent with previous reports [5,8,38,39].

### 2.3. Combination of FO and Se Counteracts the Elevation of Prosurvival Effectors in HCC827GR Cells and Restores Their Sensitivity to Gefitinib

Our previous study showed the synergistic combination effects of FO and Se on the inhibition of GRP78, β-catenin, COX-2, and the induction of CHOP in NSCLC cells [31]. Regarding the changes of these proteins between HCC827 and HCC827GR cells shown in Figure 2a, we investigated if an FO and Se combination could counteract the alterations of these protein levels caused by an adaptation to gefitinib. As expected, the combination of FO and Se increased CHOP and strikingly reduced GRP78, β-catenin, and COX-2 in HCC827GR cells (Figure 4a, left panel). As shown in the right panel of Figure 4a, the cells treated with gefitinib in combination with FO and Se had higher CHOP levels and markedly lower GRP78, β-catenin, and COX-2 levels in comparison with that treated with gefitinib alone. The elevated prosurvival effectors in HCC827GR cells were substantially reduced, and the proapoptotic CHOP was increased by the combination of FO and Se.

As a result, the gefitinib-induced apoptosis (sub-G1 fraction) in HCC827GR cells was augmented from 4.1% to 51.4% when combined with FO and Se (Figure 4b), along with the enhanced activation of caspase -3, -9 and the ER stress-related caspase-4 (Figure 4c). Accordingly, in the presence of the FO and Se combination, the viability of HCC827GR cells was decreased by gefitinib in a concentration-dependent manner, and this phenomenon was not observed in the presence of either FO or Se alone (Figure 4d). At a concentration of 1 μM, gefitinib only reduced HCC827GR cell viability to 87.4% of the control (Figure 4d). When combined with FO and Se, the cell viability was reduced to 28.3% by the same dose (1 μM) of gefitinib (Figure 4d). Though the combination of FO and Se only reduced the viability of HCC827GR cells to 73.6% (Figure 4d) and induced 27.4% of apoptosis (sub-G1 fraction) (Figure 4b), it substantially enhanced the sensitivity of HCC827GR cells to gefitinib (Figure 4b,d).

### 2.4. Combination of FO and Se Suppresses the EMT and Cancer Stem Cell Traits in Gefitinib-Resistant Subline HCC827GR

After observing the elevated EMT and cancer stem cell features of HCC827GR cells shown in Figure 2 and Figure 3, we next examined if the FO and Se combination also diminished these features when reversing their resistance to gefitinib. As shown in Figure 5a, either with or without the presence of gefitinib (1 μM), the combination of FO and Se decreased the mesenchymal (vimentin and N-cadherin) markers and increased the epithelial (E-cadherin) marker of HCC827GR cells, along with the inhibition of the amplified bypass signaling pathway molecules such as AXL and p-MET. Similarly, the lung cancer stem markers ABCG2, CD133, and CD44 were also suppressed by the combination of FO and Se (Figure 5b). Intriguingly, gefitinib (1 μM) further increased the elevated ABCG2 in HCC827GR cells, and this increase of ABCG2 was diminished by the FO and Se combination (Figure 5b).

In accordance with this, gefitinib (1 μM) increased the percentage of the cancer stem-like side population from 2.11% to 4.13% in HCC827GR cells (Figure 6). In the presence of the FO and Se combination, the side population percentages in the control and gefitinib (1 μM)-treated HCC827GR cells were reduced to 0.65% and 0.93% from 2.11% and 4.13%, respectively (Figure 6). The FO and Se combination not only exerted opposing regulatory effects on the prosurvival and proapoptotic effectors in the HCC827GR cells but also diminished their EMT features and cancer stem cell traits when restoring their sensitivity to gefitinib.

## 3. Discussion

The acquired resistance to EGFR-TKIs greatly hampers the success of targeted therapy against lung cancer. Nowadays, lots of ongoing efforts are being made to develop further generation drugs or combination strategies for overcoming this critical obstacle. A recent breakthrough was the development of third-generation EGFR-TKIs, which irreversibly block the T790M mutant EGFR, the major mechanism of acquired resistance to first-generation EGFR-TKIs [6]. However, it is fully predicted that resistance will also occur to this class of EGFR-TKIs [6]. Several mechanisms of acquired resistance to third-generation EGFR-TKIs such as rociletinib (CO-1686) and osimertinib (AZD9291) have been described in vitro and in the clinical setting [6]. In addition to developing further generation drugs targeting the secondary mutation of EGFR, other alternative strategies are imperatively needed to combat the acquired resistance.

Besides the T790M mutation, different mechanisms of acquired resistance to first-generation EGFR-TKIs have been reported [5,6]. Nonetheless, optimal treatments have not yet been clearly defined for these mechanisms of EGFR-TKI-acquired resistance [41]. An attractive strategy is the selective inhibition of the adaptive response that inhibits apoptosis and promotes the emergence of an acquired treatment-resistant phenotype [42]. It has been shown that ER stress leads to the activation of an adaptive response named the unfolded protein response [43]. Targeting the ER stress-induced unfolded protein response has been proposed to overtake cancer drug resistance [43]. Consistent with previous studies [39,44], the HCC827GR-resistant subline generated in this study does not carry the T790M mutation. In this work, we investigated the roles of prosurvival GRP78 and proapoptotic CHOP, two important ER stress response elements, in the acquired-gefitinib-resistance of HCC827GR NSCLC cells. In addition to the previously reported increase of β-catenin and COX-2 [32,33], our results demonstrated the reduced CHOP and marked elevated GRP78 in the HCC827GR compared with the parental HCC827 cells after adaptation to gefitinib. This was in agreement with the reports that the overexpression of GRP78 prevents CHOP induction to avoid apoptosis [45]. In line with our previous study showing the opposite regulation of CHOP and GRP78 by the FO and Se combination in NSCLC cells [31], this combination reduced GRP78 levels and increased CHOP protein levels in HCC827GR-resistant cells, accompanied by the restoration of their sensitivity to gefitinib. The perturbation of the apoptotic machinery has been regarded as a mechanism of acquired resistance to first-generation EGFR-TKIs [6]. The opposite regulation of CHOP and GRP78 by the FO and Se combination appears to counteract the proadaptive ER stress response that allows the HCC827GR cells to evade apoptosis upon gefitinib treatment. In consonance with our finding, a recent study by Kwon et al. reported that high GRP78 expression was an independent predictor of poor disease-free survival in EGFR-mutated lung adenocarcinoma, and it suggested the potential of the ER stress pathway as a prognostic biomarker and therapeutic target [46].

A recent study by van Lidth de Jeude et al. proposed that GRP78 may be a therapeutic target for the prevention of intestinal neoplasms without affecting normal tissue [47]. Based on their study in mice, Lee et al. described a notion that a normal organ requires only a low basal GRP78 level for function maintenance, while cancer cells require high levels of GRP78 for survival, growth, invasion, and treatment resistance [48]. In parallel, antibody targeting GRP78 was shown to enhance the efficacy of radiation in heterotopic NSCLC and glioblastoma multiforme tumor models in mice [49]. It induced apoptosis in human NSCLC (A549 and H460) and glioblastoma (U251 and D54) cell lines without affecting the viability of normal cells (MRC-5 and HUVEC) [49]. Similarly, our previous work showed the synergistic effect of FO and Se on GRP78 reduction and apoptosis induction in A549 NSCLC cells [31]. By contrast, the viability of MRC-5 (fetal lung fibroblast) cells was unsusceptible to the combination effect of FO and Se [31]. These studies suggest that targeting GRP78 for the treatment of cancer is unlikely to have major deleterious side effects [31,47,48,49]. An elevated GRP78 level has been linked to the acquired resistance of cancer cells to tamoxifen [50], 5-fluorouracil (5-FU) [51], gemcitabine [52], and targeted therapy agents such as sorafenib [53] and sunitinib [54]. Regarding the marked effect of the FO and Se combination on GRP78 reduction shown in this study, this combination might have potential in relieving other GRP78-associated treatment resistances of cancer cells.

Notably, in addition to driving drug resistance, GRP78 has been shown to promote the stemness [55,56] and EMT [57,58] of cancer cells. In agreement, our HCC827GR-resistant subline possessing an elevated GRP78 level has also raised stemness and EMT markers, such as has been reported in published research [5,38]. Based on our results, it is most likely that the elevated GRP78 in HCC827GR cells might contribute to their raised features of EMT and cancer stemness, and the diminishment of these features by the FO and Se combination might attribute to their effect on GRP78 reduction.

It has been proposed that β-catenin and COX-2 are the targets for Se to suppress the EMT and stemness traits of cancer cells [59]. Based on the substantial suppression of these two proteins by the combination of FO and Se, our previous study deduced their effect on inhibiting EMT and cancer stemness [31]. In agreement, this study showed the inhibition of EMT features and cancer stem cell traits by FO combined with Se in resistant HCC827GR cells, while the raised β-catenin and COX-2 levels were diminished by this combination. In addition to GRP78 reduction, the diminishment of β-catenin and COX-2 might also participate in the aforementioned effects of the FO and Se combination on suppressing EMT and cancer stemness.

Peculiarly, treatment with gefitinib further increased the elevated ABCG2 and side population percentage in HCC827GR cells, which also could be abolished by FO combined with Se. The mechanisms proposed above are not sufficient to explain this phenomenon. Additional action mechanisms of the FO and Se combination remain to be elucidated, and they might have potential to prevent the possible expansion of cancer stem cells by treatment after the acquisition of resistance.

Recently, it has been reported that the EMT beyond EGFR mutations per se is a common mechanism for acquired resistance to EGFR TKI [38]. Our results showed the combination effect of FO and Se on the reversal of the EMT and acquired-gefitinib-resistance in HCC827 NSCLC cells. Apart from targeting the mutated EGFR, our findings unravel an alternative potential strategy to counteract the acquired-gefitinib-resistance phenotype through the modulation of ER stress response elements, and shed light on the potential benefit of combining FO and Se for relieving the acquired resistance of NSCLC patients to targeted therapy.

The effective concentrations of FO and Se used in this study are clinically achievable. A cross-sectional study by Kuriki et al. collected the seven-day weighed diet records of Japanese dietitians and estimated their dietary intakes of DHA, EPA, and omega-3 highly unsaturated fatty acid (HUFA) derived from marine foods; they also analyzed the plasma concentrations of these fatty acids [36]. Kuriki et al. showed that the estimated mean dietary intakes of HUFA were 1092 mg in men and 971 mg in women, and their plasma concentrations of DHA and EPA were all above 200 μM [36]. On the other hand, mean plasma Se levels of 492.2 and 639.7 ng/mL were achieved in prostate cancer patients receiving 1600 and 3200 μg/day of selenized yeast, respectively, for up to 24 months [35]. The results of these studies in human subjects [35,36] might help to estimate the effective oral doses of FO and selenized yeast for combination treatment in patients. Nevertheless, for the sake of clinical trials in the future, further in vivo studies to evaluate the efficacy and safety of this combination treatment are warranted.

## 4. Materials and Methods

### 4.1. Cell Culture

The human lung adenocarcinoma cell line HCC827 and its gefitinib-resistant subline HCC827GR were kindly provided by Dr. Yu-Ting Chou, Institute of Biotechnology, National Tsing Hua University, Taiwan. The HCC827GR was derived from HCC827 cells by culturing in a medium containing escalating concentrations of gefitinib for 6 months to select cells which could grow in micromolar concentrations of gefitinib [60]. These two cell lines were maintained in a Roswell Park Memorial Institute (RPMI)-1640 medium (Gibco, Massachusetts, MA, USA) supplemented with 10% fetal bovine serum (Corning, NY, USA) and 1 × penicillin-streptomycin-glutamine (Corning). Cells were cultured at 37 °C in a water-jacketed 5% CO_2_ incubator.

### 4.2. Reagents and Chemicals

Se yeast was chosen as the form of Se to treat NSCLC cells. The stock solutions of Se yeast and FO (each gram contained 220 mg of DHA and 330 mg of EPA) were provided by Dr. Chih-Hung Guo (Institute of Biomedical Nutrition, Hung-Kuang University, Taichung, Taiwan). They were then aliquoted and stored at −20 °C (FO) and −80 °C (Se yeast). Both of them were diluted in a sterile culture medium immediately prior to use. The concentration of FO mentioned in the text represents its content of omega-3 fatty acid (DHA and EPA). Gefitinib (Iressa), sulforhodamine B (SRB), trichloroacetic acid, and propidium iodide were purchased from Sigma (St. Louis, MO, USA). DCV dye was from Invitrogen (Invitrogen, Carlsbad, CA, USA).

### 4.3. Measurement of Cell Viability

The cells were seeded in 96-well plate (2000 cells/well) for 24 h and then treated with drugs or a sterile culture medium (control) for 72 or 96 h, as indicated in legends. The cell viability was measured with an SRB colorimetric assay. Briefly, the cells were fixed with 10% trichloroacetic acid and incubated for 1 h at 4 °C. The plates were then washed twice with tap water and air dried. The dried plates were stained with 80 μL of 0.4% (*w/v*) SRB prepared in 1% (*v/v*) acetic acid for 30 min at room temperature. The plates were rinsed quickly twice with 1% acetic acid to remove unbound SRB, and then they were air dried until no moisture was visible. The bound dye was solubilized in 20 mmol/L Tris base (200 μL/well) for 5 min on a shaker. Optical densities were read on a microplate reader (ELx800; BioTek, Winooski, VT, USA) at 570 nm. The optical density was directly proportional to the viable cell number over a wide range.

### 4.4. Analysis of Apoptotic Sub-G1 Fraction by Propidium Iodide Staining

One day after being seeded in 10-cm dish (3.5 × 10^5^ cells/dish), the cells were treated with agents, as indicated in the figure for 96 h. At harvest, cells were fixed in ice-cold 70% ethanol and stored at −20 °C. Cells were then washed twice with ice-cold phosphate-buffered saline (PBS) and then incubated with RNase and the DNA-intercalating dye propidium iodide (50 μg/mL) at room temperature for 20 min. The percentages of the apoptotic sub-G1 fraction were then analyzed using a CytoFLEX flow cytometer (Beckman Coulter, Inc., Indianapolis, IN, USA). A minimum of 10,000 events were collected and analyzed.

### 4.5. Side Population Analysis by DyeCycle Violet (DCV) Exclusion

One day after being seeded in 10-cm dish (3.5 × 10^5^ cells/dish), the cells were treated with agents, as indicated in the figure, for 72 h. Then, the cells were removed from the culture dish with 0.25% trypsin and 0.05% ethylenediaminetetraacetic acid (EDTA), washed with phosphate-buffered saline (PBS), and resuspended at a concentration of 1 × 10^6^ cells/mL in RPMI 1640 media containing 10% fetal bovine serum (FBS) at 37 °C. These (1 mL, 1 × 10^6^) cells were then incubated at 37 °C for 90 min with either 1 μL (final stain concentration is 5 μM) of DCV alone or in combination with reserpine (10 μM in HCC827 and 5 μM in HCC827GR) or 200 μM verapamil under light protection. Subsequently, cells were centrifuged immediately for 5 min at 240× *g* and 4 °C, and then they were resuspended in ice-cold PBS; the propidium iodide (final stain concentration was 50 μg/mL) was then added to discriminate dead cells. The cells were then maintained at 4 °C until analyzed by the CytoFLEX flow cytometer (Beckman Coulter) at an excitation of 405 nm and emissions of 450 and 660 nm. The ATP-binding cassette (ABC) transporter inhibitors reserpine and verapamil were used to conform the gating of side population, as indicated.

### 4.6. Western Blot

After being seeded in 10-cm dishes at a density of 3.5 × 10^5^ cells/dish for 24 h, cells were treated with agents, as described in the figure, for 72 h. On the day of harvest, cells were scraped with a rubber policeman, and the whole-cell lysates were extracted with 1 × radioimmunoprecipitation lysis buffer (Millipore, Billerica, MA, USA) containing 1 × protease inhibitor cocktail, full range (FC0070-0001, BIONOVAS, Toronto, Canada), 1 × tyrosine phosphatase inhibitor cocktail (FC0020-0001; BIONOVAS), and 1 × serine/threonine phosphatase inhibitor cocktail (FC0030-0001, BIONOVAS). The protein extracts were resolved by sodium dodecyl sulfate–polyacrylamide gel electrophoresis and subsequently transferred to a polyvinylidene difluoride membrane (GE Healthcare, Pittsburgh, PA, USA) by electroblotting. The membranes were blocked with 5% bovine serum albumin in a TBST buffer (Tris-buffered saline with Tween 20, 25 mM Tris-HCl, 125 mM NaCl, and 0.1% Tween 20) for 1 h at room temperature and incubated with primary antibody overnight at 4 °C and then with horseradish peroxidase-conjugated secondary antibody for 1 h at room temperature. An intensive wash with TBST buffer was performed after each incubation. The immune complexes were visualized using enhanced Chemiluminescence (ECL) Reagent Plus (Perkin Elmer, Waltham, MA, USA) on the Syngene G: Box chemi XL gel documentation system (Syngene, Cambridge, UK) according to the manufacturer’s instructions. The quantification of the Western blot band intensities was performed using the ImageJ software (ImageJ bundled with 64-bit Java 1.8.0_112, National Institutes of Health, Bethesda, MD, USA) downloaded from https://imagej.nih.gov/ij/download.html.

### 4.7. Antibodies

Primary antibodies against, non-phospho (active) β-Catenin (Ser33/37/Thr41, #8814), CHOP (#2895), AXL (#8661), ABCG2 (#4477), E-cadherin (#3195), cleaved caspase-9 (Asp315, #9505), and cleaved caspase-3 (Asp175, #9664) were purchased from Cell Signaling (Danvers, MA, USA). Primary antibodies for CD133 (ab19898), CD44 (ab51037), cleaved caspase-4 (ab75182), COX-2 (ab62331), vimentin (ab92547), N-cadherin (ab76011), and glyceraldehyde-3-phosphate dehydrogenase (GAPDH) (ab8245) were purchased from Abcam (Cambridge, MA, USA). The primary antibody for GRP78 (#PA1-014A) was purchased from Thermo Fisher Scientific™ (Waltham, MA, USA). The primary antibody for p-MET (sc-34086) was obtained from Santa Cruz Biotechnology (Santa Cruz, CA, USA).

### 4.8. Statistical Analysis

Cell viability data are expressed as mean ± standard error. Differences between the cell viabilities of the control and treated groups were evaluated by a one-way ANOVA followed by Dunnett’s *t* test. A probability value of *p* < 0.05 was considered statistically significant. A single asterisk (*) indicates *p* < 0.05, double asterisks (**) indicate *p* < 0.01, and triple asterisks (***) indicate *p* < 0.001.

## 5. Conclusions

As shown in Scheme 1, our previous study showed the opposite regulation of CHOP and GRP78 and the suppression of β-catenin and COX-2 by FO combined with Se in NSCLC cells [31]. Through modulating these targets, the FO and Se combination appears to counteract the adaptive response effect in HCC827GR cells, which endows the acquisition of the gefitinib-resistance phenotype, including the raised features of EMT and cancer stemness. Though the further detailed mechanistic interactions between the aforementioned molecular events remain to be elucidated, our results suggest the potential use of combining FO and Se in relieving the acquired resistance of cancer cells to gefitinib or other treatment modalities.

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
