# Peer review of "Omega-3 Fatty Acid-Enriched Fish Oil and Selenium Combination Modulates Endoplasmic Reticulum Stress Response Elements and Reverses Acquired Gefitinib Resistance in HCC827 Lung Adenocarcinoma Cells"

_marinedrugs, 2020, doi:10.3390/md18080399_

Round 1

Reviewer 1 Report

A clear, concise and well-written manuscript that presents no major flaws. Just one minor point: the SRB colorimetric assay measures cell viability, not proliferation. Accordingly, any references to cell proliferation in the text and Figs. should be eliminate; the words "ceel viability" should be used.

Author Response

Response: We thank the reviewer's carefully reviewing of our manuscript and appreciate the insightful comments. We have revised our manuscript according to reviewer's comments and highlighted the changes in red.

Reviewer 2 Report

In this report, the authors established HCC827GR cells resistant to the anticancer drug gefitinib. They confirmed that these cells were resistant to gefitinib, and confirmed the strong induction of the molecular chaperone GRP78 and the suppression of CHOP expression. Moreover, ABCG2 was strongly induced, and side-population analysis showed that ABCG2 was activated in HCC827GR cells. Here, they showed the combination of fish oil and selenium canceled the gefitinib resistance of HCC827GR cells. Furthermore, the change in the expression level characteristic of HCC827GR cells was canceled. It also canceled the ABCG2 activity of HCC827GR cells. These show that the combination of fish oil and selenium is useful for anticancer drug treatment using gefitinib.

Please consider adding the following to the discussion:

If the combination of fish oil and selenium is actually used for treatment, is the treatment time and concentration appropriate for the treatment? In this paper, the authors observed the drug treatment for 3 days, but did they observe each data on the time course? In addition, what methods are considered to be effective in introducing them to patients? How is an effective dose of the drug delivered as a result of the pharmacokinetic, such as an intravenous injection? Furthermore, could you add an explanation for the previous data about the lack of effect on normal cells?

Minor Points

In the abstract, the author should describe EGFR and EMT abbreviations.

In Figure 2, the bands for cox-2, Vimentin, and AXL in HCC827 cells are completely invisible, so the author should increase the intensity so that the bands are visible in the printed paper.

Author Response

Comments and Suggestions for Authors

In this report, the authors established HCC827GR cells resistant to the anticancer drug gefitinib. They confirmed that these cells were resistant to gefitinib, and confirmed the strong induction of the molecular chaperone GRP78 and the suppression of CHOP expression. Moreover, ABCG2 was strongly induced, and side-population analysis showed that ABCG2 was activated in HCC827GR cells. Here, they showed the combination of fish oil and selenium canceled the gefitinib resistance of HCC827GR cells. Furthermore, the change in the expression level characteristic of HCC827GR cells was canceled. It also canceled the ABCG2 activity of HCC827GR cells. These show that the combination of fish oil and selenium is useful for anticancer drug treatment using gefitinib.

Response: We thank the reviewer's carefully reviewing of our manuscript and appreciate the insightful comments and helpful suggestions. We have revised our manuscript according to reviewer's comments and highlighted the changes in red. In the following, we list our response to the comments point by point.

Please consider adding the following to the discussion:

If the combination of fish oil and selenium is actually used for treatment, is the treatment time and concentration appropriate for the treatment? In this paper, the authors observed the drug treatment for 3 days, but did they observe each data on the time course? In addition, what methods are considered to be effective in introducing them to patients? How is an effective dose of the drug delivered as a result of the pharmacokinetic, such as an intravenous injection?

Response: Thanks for the reviewer’s helpful suggestion. Our unpublished data show that the gefitinib-sensitizing effect of FO and Se combination in HCC827GR cells is more profound at 96 h than 72 h. The longer time intervals of treatment might be more appropriate. To estimate the effective doses of FO and selenized yeast for combination treatment in patients, we have cited the references reporting human plasma levels of DHA, EPA and Se achieved by daily intake of omega-3 highly unsaturated fatty acid (HUFA) derived from marine foods and selenized yeast, respectively. Please see 11, line 316-327 in our revised version. We have revised our discussion according to your insightful suggestion.

Furthermore, could you add an explanation for the previous data about the lack of effect on normal cells?

Response: Thanks for the reviewer’s helpful suggestion. Our previous data showed profound effect of FO and Se combination on the reduction of GRP78 in A549 lung cancer cells. The lack of effect on normal cells might attribute to that normal organ requires only a low basal GRP78 level for function maintenance, while cancer cells require high levels of GRP78 for survival, growth, invasion and treatment resistance (Sci Rep 2017, 7, 40919, doi:10.1038/srep40919.). In parallel, it had been shown that antibody targeting GRP78 induced apoptosis in human NSCLC (A549 and H460) and glioblastoma (U251, and D54) cell lines without affecting the viability of normal cells (MRC-5 and HUVEC)( Clin Cancer Res 2017, 23, 2556-2564, doi:10.1158/1078-0432.CCR-16-1935.). We have cited these references and revised our discussion accordingly. Please see page 10, line 274-283 in our revised version.

Minor Points

In the abstract, the author should describe EGFR and EMT abbreviations.

Response: Thanks for the reviewer’s carefully reviewing. We have revised our abstract accordingly and highlighted the changes in red.

In Figure 2, the bands for cox-2, Vimentin, and AXL in HCC827 cells are completely invisible, so the author should increase the intensity so that the bands are visible in the printed paper.

Response: Thanks for the reviewer’s helpful suggestion. We have revised our Figure 2 accordingly.

Reviewer 3 Report

The paper is of interest and well written. I have no particular comments to add. 

Author Response

Comments and Suggestions for Authors

The paper is of interest and well written. I have no particular comments to add.

Response: We thank the reviewer's carefully reviewing of our manuscript and really appreciate the warm comments.
